# Correlative single-cell hard X-ray computed tomography and X-ray fluorescence imaging
Zihan Lin[1,6], Xiao Zhang [1,6], Purbasha Nandi[1], Yuewei Lin [2], Liguo Wang[3], Yong S. Chu[4], Timothy Paape[1,5], Yang Yang [4] ✉, Xianghui Xiao [4] ✉ & Qun Liu [1,4] ✉

X-ray computed tomography (XCT) and X-ray fluorescence (XRF) imaging are two non-invasive imaging techniques to study cellular structures and chemical element distributions, respectively. However, correlative X-ray computed tomography and fluorescence imaging for the same cell have yet to be routinely realized due to challenges in sample preparation and X-ray radiation damage. Here we report an integrated experimental and computational workflow for achieving correlative multi-modality X-ray imaging of a single cell. The method consists of the preparation of radiation-resistant single-cell samples using live-cell imaging-assisted chemical fixation and freeze-drying procedures, targeting and labeling cells for correlative XCT and XRF measurement, and computational reconstruction of the correlative and multi-modality images. With XCT, cellular structures including the overall structure and intracellular organelles are visualized, while XRF imaging reveals the distribution of multiple chemical elements within the same cell. Our correlative method demonstrates the feasibility and broad applicability of using X-rays to understand cellular structures and the roles of chemical elements and related proteins in signaling and other biological processes.

Cells are the building blocks of living organisms. X-rays have deep-penetration power and are ideal for non-invasive imaging of cells at nanometer resolutions[1,2]. Transmission X-ray computed tomography (XCT) and X-ray scanning fluorescence (XRF) imaging are two nanometer-resolution imaging techniques capable of studying cellular structure and function in high spatial resolution. XCT is analog in its configuration to optical bright-field microscopy, utilizing a condenser to focus X-rays onto the sample and employing a zone plate as an X-ray objective lens to magnify images for full-field tomographic imaging[3,4]. Meanwhile, XRF imaging operates as a spectroscopic technique based on scanning, utilizing element-specific X-ray fluorescence spectra to map multiple chemical elements[5–7]. The two methods complement each other, where XCT provides the overall cellular and subcellular structures while XRF imaging generates elemental maps crucial for understanding element-related uptake, metabolism, and cellular localization.

Both X-ray imaging modalities have found extensive applications in biological research[3,5,8]. XCT utilizes either absorption or phase contrast for imaging biological cells. Soft X-rays at energy within the water window (183–530 eV) exhibit a larger mass absorption cross-section difference for carbon and nitrogen relative to water, thus creating absorption contrast for tomographic imaging of hydrated cells[1,9]. However, these soft X-rays are also associated with high parasitic X-ray absorption and radiation damage, necessitating the use of cryofixation and data collection at cryogenic temperatures[9]. Soft X-rays are also limited to a short focal length and samples thinner than 20 μm, and thus, large cells, such as some plant and animal cells, may be challenging for soft X-ray computed tomography. In comparison, hard X-rays with an energy of 5 keV or higher possess a smaller cross-section but generate much less radiation damage while penetrating to great deeper, e.g., 1 mm for 10 keV X-rays[10]. Hard X-rays have been utilized to study cellular structures of unstained MC3 preosteoblasts[11] and fission yeast cells[12]. Due to the small cross-section difference between carbon, nitrogen, and water, the absorption contrast is very low for imaging biological cells. Hence, previous researchers preferred to use phase contrast mechanisms by a Zernike phase plate to create a phase delay between absorbed and scattered X-rays[11,12]. To enhance the contrast for hard X-ray imaging, staining cells using heavy metals such as osmium, lead, and uranyl

[1]Biology Department, Brookhaven National Laboratory, Upton, NY 11973, USA. [2]Computational Science Initiative, Brookhaven National Laboratory, Upton, NY 11973, USA. [3]Laboratory for BioMolecular Structure, Brookhaven National Laboratory, Upton, NY 11973, USA. [4]National Synchrotron Light Source II, Brookhaven National Laboratory, Upton, NY 11973, USA. [5]U.S. Department of Agriculture's Agricultural Research Service at Children's Nutrition Research Center, Houston, TX 77030, USA. [6]These authors contributed equally: Zihan Lin, Xiao Zhang. ✉e-mail: yyang@bnl.gov; xiao@bnl.gov; qunliu@bnl.gov

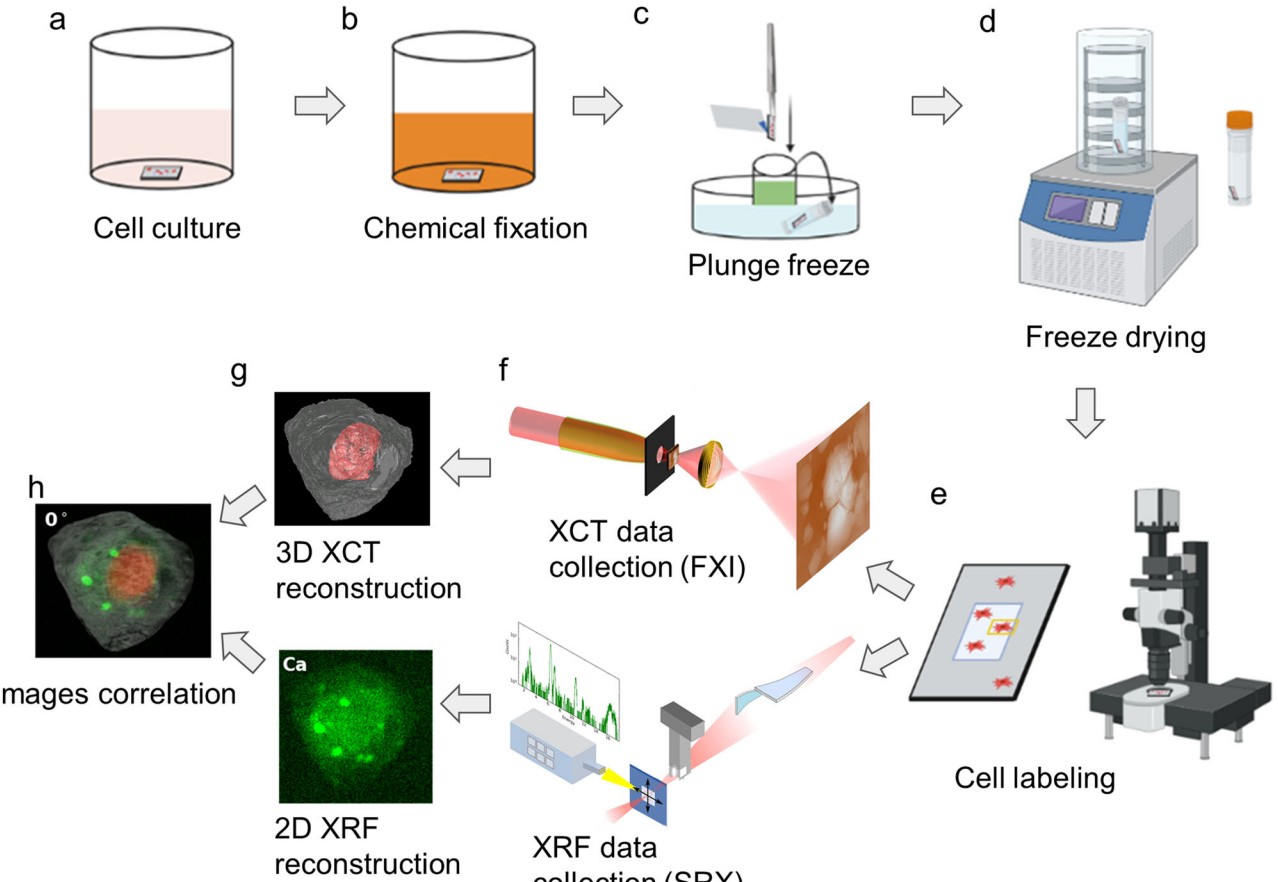

**Fig. 1 | Overall workflow of single-cell correlative X-ray imaging. a** HEK293T cells are grown adherently on the flat surface of a poly-lysine-treated $Si_3N_4$ membrane. **b** Adherent cells are washed twice with PBS, and a chemical fixation solution is added to fix the cell. **c** $Si_3N_4$ membrane is washed twice with 150 mM ammonium acetate and then plunge-freezed using a Vitrobot. The membrane is stored in a screw-cap microcentrifuge tube. **d** The tube with the sample is transferred to a freeze-dryer device for freezing-drying. **e** Cells are labeled using a confocal microscope for multiple beamline data collection. **f** Single-cell XCT and XRF data are collected using X-ray computed tomography at the FXI beamline and X-ray fluorescence micro-scopy at the SRX beamline, respectively. **g** Reconstruction of 3D XCT tomograms and 2D XRF fluorescence map. **h** Correlation of multi-modality images.

is generally required, but such a method limited the resolution and caused stain-related artifacts[12].

XRF is highly sensitive and capable of detecting elemental distributions in a sample at sub-ppm (parts per million) levels. Unlike other methods, XRF does not require the use of external markers or tags, enabling simultaneous detection of multiple elements with exceptional resolution using a highly focused X-ray beam[5,6]. This characteristic makes XRF suitable to probe and differentiate compositions of both metal and non-metal elements inside cells, thereby aiding in the understanding of their roles in signaling, homeostasis, and other cellular functions. As hard X-rays can detect biological elements with an atomic number of 15 (phosphorus) or higher and are insensitive to water, both hydrated and dehydrated samples have been employed for XRF[13–15]. XRF has been utilized to study cellular functions of metals, for example, zinc in leaf cells[16], copper in neuronal cells[17], and calcium, phosphorus, and iron in algae[18].

Although it is attractive to image the same cells using multiple X-ray modalities[19,20], challenges exist in developing suitable sample preparation protocols to optimize contrast and mitigate the radiation damage associated with the measurement of the same cells, as well as data correlation between the two imaging modalities. Addressing these challenges is crucial to facilitate the routine application of these correlative techniques for precise imaging at cellular and subcellular levels. Here we developed a single-cell X-ray imaging workflow that encompasses sample preparation, targeting, multi-modality data acquisition, and computational alignments of multi-modality image data at nanometer resolutions. Our method offers a reliable and practical approach for obtaining the structure and chemical information of a single cell with nanometer-level resolution. The results of our research pave the way for the broad application of hard X-rays for correlative characterization of cell structure and function, opening up new possibilities for advanced biological investigations.

## Results

### A single-cell multi-modality X-ray imaging workflow

HEK293 cells were utilized in this study to demonstrate the workflow. Cells are fragile and their manipulation and targeting require the use of compatible sample holders and experimental setups across various imaging modalities. Figure 1 depicts the devised workflow for achieving single-cell-level correlative multi-modal imaging. To begin, an X-ray transparent $Si_3N_4$ membrane is used for growing cells (Fig. 1a). Subsequently, the grown cells are chemically fixed using paraformaldehyde (PFA) to preserve cellular structure and morphology (Fig. 1b). Fixed cells are manually blotted using filter paper and rapidly plunge-freeze into liquid ethane using a Vitrobot (Thermo Fisher Scientific Inc.), and later are transferred into liquid nitrogen (Fig. 1c). The vitrified cells were then freeze-dried[21] to retain cell structure while gradually depleting water through sublimation (Fig. 1d). Dried cells are imaged using a confocal light microscope for targeting and labeling cells (Fig. 1e). The labeled cells are measured first with XCT and then with XRF (Fig. 1f). XCT and XRF data are processed separately to obtain respective 3D tomograms and 2D chemical maps (Fig. 1g). Two sets of images are then computationally aligned to generate the final integrated images (Fig. 1h).

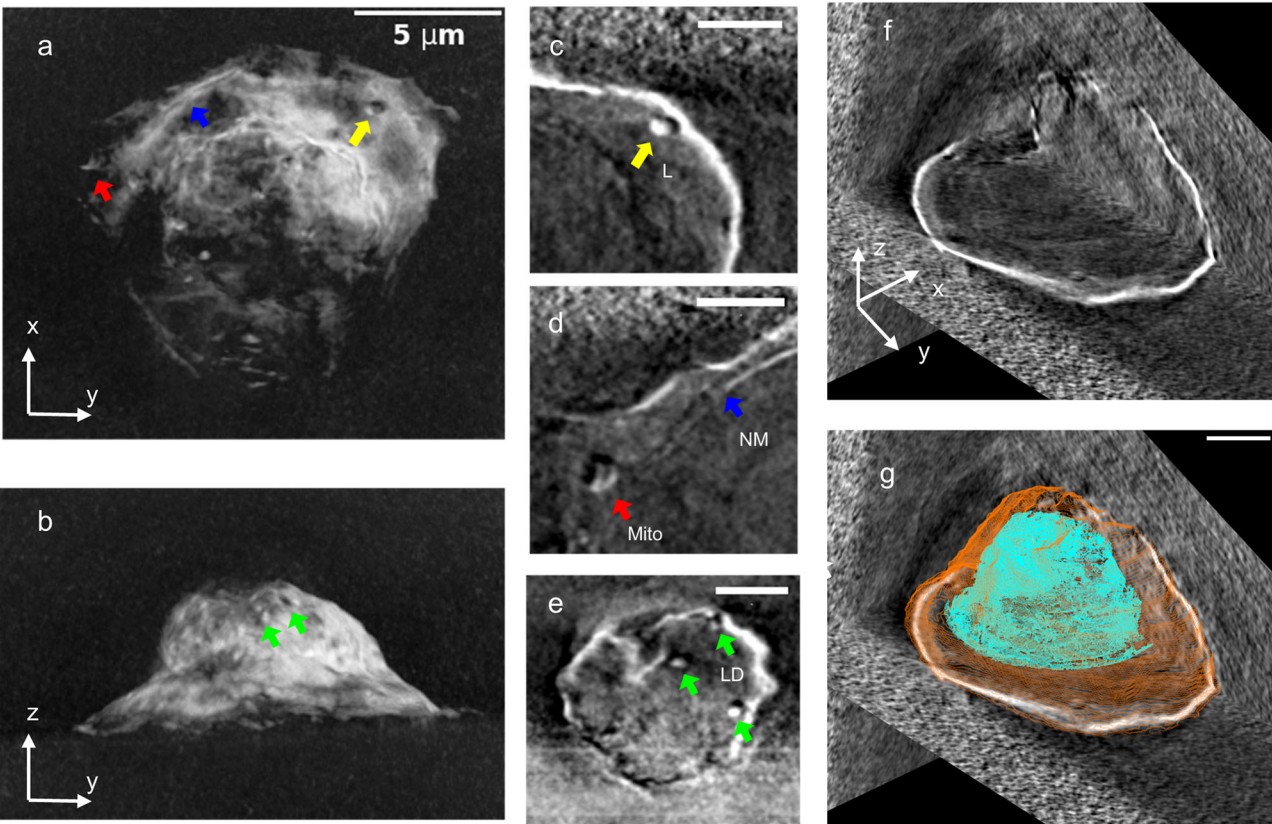

**Fig. 2 | Single-cell X-ray computed tomography. a–b** Total projection of the reconstructed X-ray tomogram in XY (**a**) and YZ (**b**) directions. **c–e** Cellular structure features including mitochondrion (**c**), nucleus membrane (**d**), and lipid droplets (**e**). The diameter for lysosome (L, yellow), mitochondria (Mito, red), and lipid droplets (LD, green) are 500 nm, 800 nm, and 350 nm, respectively. **f** Orthoslices of the cell along three directions. The whole cell is visualized in 3D (XY, YZ, and XZ) views. **g** View of (**f**) superimposed with the segmentation for nucleus (cyan) and plasma membrane (orange). The scale bar is 5 μm for (**a**, **b**), and 2 μm for (**c–g**).

This integrated workflow provides a comprehensive and detailed visualization of the cellular structure and the distribution of chemical elements within the same cell.

### Single-cell sample preparation and labeling

Freeze drying plays a critical role in preparing cell samples for XRF experiments[22,23]. However, XRF imaging is limited to detecting fluorescence-emitting chemicals such as Zn and Ca while does not provide information about the carbon-, nitrogen-, and oxygen-rich cellular structure. Consequently, fixation conditions optimized for XRF imaging may not be optimal for the correlated XCT, which aims to reveal the three-dimensional cellular structure. To address this issue, the fixation conditions were optimized using 4% paraformaldehyde (PFA), a routinely employed fixative for immunolabeling and X-ray fluorescence imaging[23,24]. The goal was to strike a balance between maintaining the integrity of the cellular structure for XCT and preserving chemical elements for subsequent XRF imaging. Upon the addition of 4% PFA to the cultured cells, the entire fixation process was continuously monitored using a Nanolive imaging system at 30-s intervals over two hours (Supplementary Movie 1). The snapshots before and after fixation are shown in Supplementary Fig. 1. No apparent changes were observed in the fixed cells compared to the initial time point (0 min). Furthermore, we did not observe membrane blebbing or loss of cytoplasmic content during the entire 2-h fixation period[25].

Chemical fixation of cells is essential to maintain cellular structures for XCT analysis. Insufficient PFA fixation (0 min and 15 min) resulted in bubbles and the loss of internal features during XCT analysis (Supplementary Fig. 2a, b). Without fixation (0 minutes), the freeze-dried cell collapsed and was flatted on the Si₃N₄ membrane (Supplementary Fig. 2a).

Sixty-minute fixation did not result in bubbles, but the internal cellular features were mostly lost compared to those of the 120-min fixation (Supplementary Fig. 2c and Fig. 2).

To perform correlative XCT and XRF imaging on a single cell, the cell density on the Si₃N₄ membrane was optimized to ensure sufficient separation between cells. Additionally, a confocal microscope was used to capture both the overall view and zoomed-in images of cells on the Si₃N₄ membrane (Supplementary Fig. 3). The cells were labeled, and the labeled locations were utilized to target the same cells in XCT and XRF experiments. This multi-resolution imaging approach can be employed to label and target any cells on a Si₃N₄ membrane.

### Hard X-ray computed tomography

XCT experiments were conducted at the full-field X-ray Imaging (FXI) beamline at the National Synchrotron Light Source II (NSLS-II) at Brookhaven National Laboratory (BNL). The FXI beamline is equipped with a Transmission X-ray Microscope (TXM) capable of rapid measurements at 30 nm spatial resolution in less than 1 min[26]. To minimize X-ray radiation damage to the samples, an optical telescope, coaligned with the X-ray beam, was utilized for target cell identification and pre-alignment. This telescope ensured that the samples were not exposed to the X-ray beam until the actual XCT data collection started. Lower energy X-rays increase intensity and contrast. With the optics set up at the FXI beamline, 7 keV is the lowest X-ray energy we can reach. We conducted all the tomography experiments at 7 keV.

The tomograms of the target cells were reconstructed using TXM sandbox[27] and visualized with IMOD[28]. Figure 2a, b displays the projection of a cell's tomogram, revealing a well-defined plasma membrane encompassing rich cellular structures. The shape of the cell matches the confocal

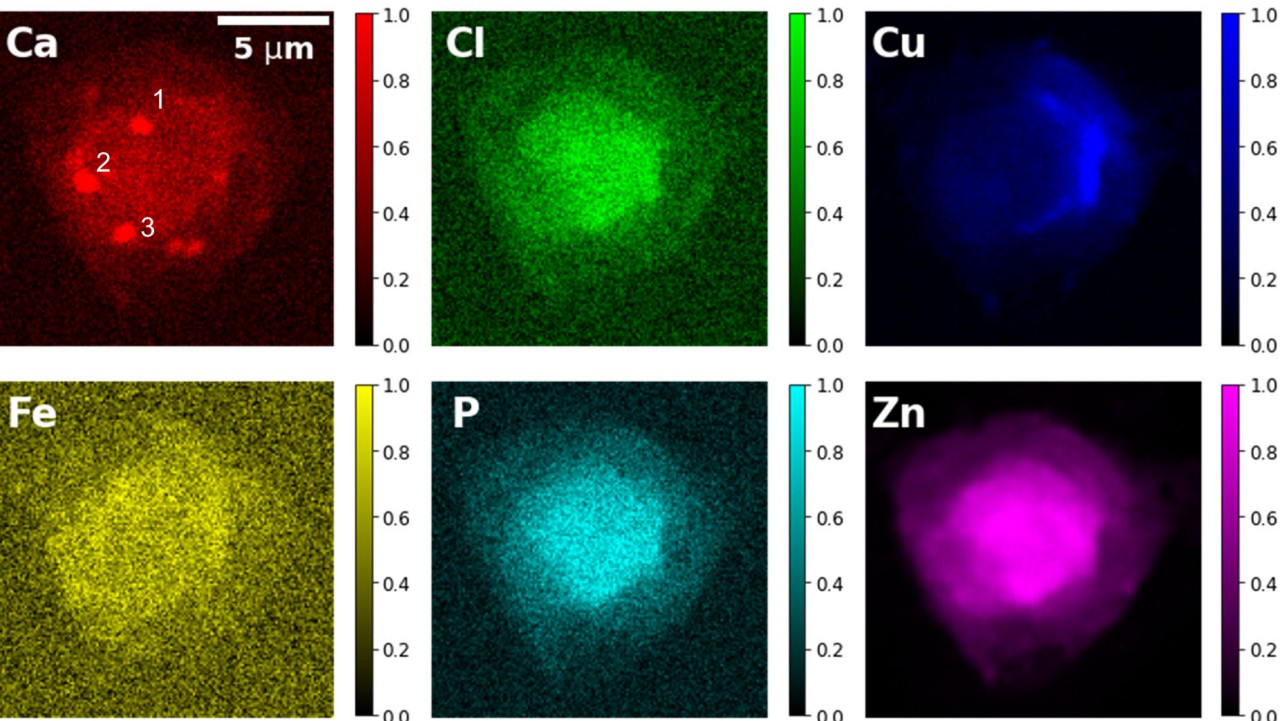

**Fig. 3 | X-ray fluorescence mapping of multiple chemical elements.** The X-ray emission energy is 12 keV. Intensities are normalized for each component. The sizes of the three calcium-rich regions (labeled with numbers 1–3) are 1.0 μm, 1.1 μm, and 0.9 μm, respectively. The scale bar is 5 μm.

microscope image depicted in Supplementary Fig. 3. Within the cell, two elongated organelles of about 500–800 nm in size are visible close to the plasma membrane (Fig. 2c, d). Based on their appearance, these organelles are likely bilayer endosome/lysosome (Fig. 2c) or multilayer mitochondrion (Fig. 2d). In addition, we observed double membrane structures (Fig. 2d), i.e., plasma membrane and intracellular membrane. By considering their length and smooth curvature of the internal membrane, we assigned it as the nucleus membrane. Upon viewing the cell from the top of the $Si_3N_4$ membrane, a few small vesicles of 300 nm can be identified (Fig. 2e). Based on the size and high X-ray contrast, we assigned these vesicles as lipid droplets. These lipid droplets exhibit spherical shapes, contrasting with the surrounding air. Another compartment membrane structure is displayed between the plasma membrane and intracellular membrane (Fig. 2e). However, we were not able to rationally assign the membrane structure to any known organelle. Based on the features of the nucleus membrane and high contrast from chromatin, we were able to segment the nucleus manually inside the cell (Fig. 2f, g). The 2D views of the 3D volume along three orthogonal planes demonstrate the positioning of the segmented nucleus within the cell (Fig. 2g). More structure details of this cell can be visualized in Supplementary Fig. 4 and Supplementary Movie 2.

## X-ray fluorescence imaging

XCT is an effective method for visualizing cellular structures. Nonetheless, due to low abundance, it cannot reveal the chemical element of trace micronutrient metals such as Zn, Fe, Cu, and Ca, essential for proper cell function. To investigate the elemental distributions within the same cell, we performed XRF measurement at the submicron resolution X-ray spectroscopy (SRX) beamline at NSLS-II. In order to precisely locate the same cell, we utilized an in-line optical microscope at SRX to identify and center the cell for the collection of raster scanning X-ray fluorescence data. To achieve high spatial resolution, we utilized a beam size of approximately 100 nm and a step size of 100 nm at an X-ray energy of 12 keV.

Figure 3 presents the chemical mapping of six common chemical elements: Ca, Cl, Cu, Fe, P, and Zn. Among these, zinc was predominantly localized in the center of the cell, i.e., the nucleus, with its concentration

gradually decreasing but remaining within the nucleus region. Phosphorus is mostly enriched in the chromatin/nucleus, consistent with its co-localization with zinc, which is known to bind to transcription factors to regulate gene transcription in the nucleus[29]. Interestingly, calcium appears to be distributed in separate regions of various sizes of about 1 μm or smaller in diameter. Given calcium's cytotoxic nature, it is typically confined to calcium stores[30]. Therefore, it is likely that these calcium-rich regions correspond to calcium stores associated with ER or mitochondria. Copper seems to be attached to the outside of the phosphorus and zinc map but extends beyond the zinc map (Supplementary Fig. 5). The chloride distribution appears to resemble that of zinc and phosphorus, suggesting its localization to the nucleus. Unlike other elements, which remain immobile during the fixation process, chloride is soluble within the cell. It remains unclear why it remained in the nucleus despite the PFA fixation.

## Correlation of X-ray computed tomography and fluorescence data

We established a correlation between the 3D tomography image and the 2D fluorescence data obtained from the same cell, displaying chemical element localization within the cell. To achieve accurate alignment between both images, additional XRF data were acquired by performing rotations at −15°, 25°, and 55° along the z-axis, consistent with the rotation axis employed at the FXI. To generate correlation maps, the 3D tomogram was appropriately rotated to match the orientation of the 2D fluorescence maps. Subsequently, we performed the total projections of the segmented cell plasma membrane and nucleus onto their respective 2D fluorescence maps (Fig. 4 and Supplementary Fig. 6). These multi-orientation XRF data facilitated the precise overlay of fluorescence signals onto the corresponding cellular structures, enabling the visualization and examination of chemical element distribution within the cell. For instance, the four orientations utilized for the calcium maps indicate that these calcium stores are situated close to, yet outside of, the nucleus, suggesting their association with the rough ER or mitochondria. Interestingly, copper is accumulated at one side of the cell and is also close to, but outside of, the nucleus. The elongated shape of the Cu map along the nucleus membrane suggests its localization to rough ER or Golgi apparatus. Copper and calcium have been reported to be involved in the function of

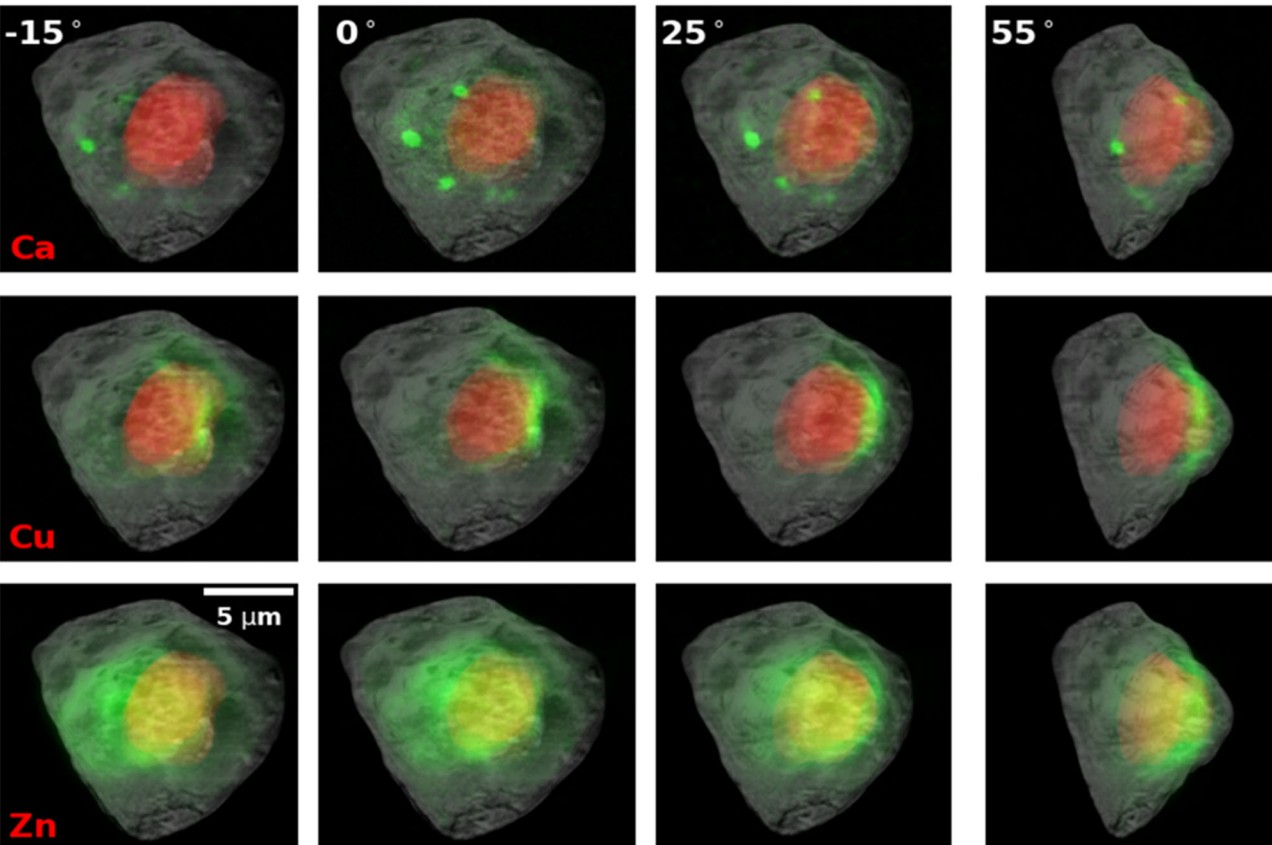

**Fig. 4 | Correlation of multi-modality data by superimposition of 2D XRF and 3D XCT images along 4 different view angles.** The cell was rotated along the *vertical* axis by −15°, 0°, 25°, and 55°, respectively. Gray scale maps show the max projection of the 3D X-ray tomogram of the cells. Red masks show a segmented nucleus from the tomogram. Green maps show the corresponding localization of multiple elements. The scale bar is 5 μm.

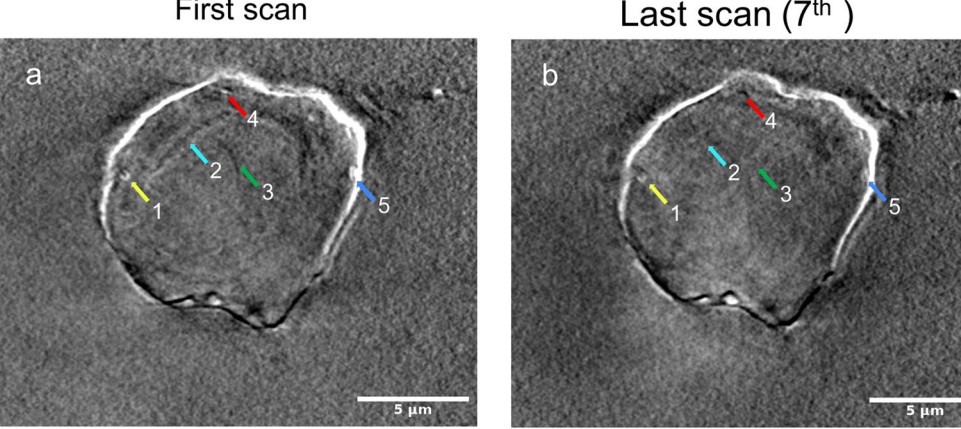

**Fig. 5 | Radiation damage in X-ray tomographic imaging.** Representative slices of the reconstruction results of (**a**) from the first X-ray computed tomography scan and (**b**) from the last scan (7th scan). The same experimental conditions are used for the first the last scans. Due to radiation damage, features indicated by color arrows in (**a**) were weakened (1,5) or disappeared (2–4) in (**b**). Scale bars are 5 μm.

both mitochondria and the ER, raising intriguing questions for future studies[31,32]. The correlative and non-invasive multi-modality imaging of the same cell provides a tool to further characterize element distribution inside a cell which may not be possible using other imaging tools.

### Radiation damage

To assess the X-ray-caused radiation damage, a series of six consecutive XCT scans were performed on the same cell, employing different experimental conditions such as exposure time, rotation speed, and X-ray energy.

Subsequently, a 7th scan was conducted with the same parameters as the initial one. The resulting tomograms were reconstructed using TXM sandbox and IMOD, employing the same reconstruction parameters and post-processing filters. The first and the last tomograms were chosen for comparative analysis (Fig. 5). The results indicate that the repeated tomography data collection induced the cell-volume shrinkage by approximately 3-11 % along each dimension (Supplementary Fig. 7). In addition, high-resolution features seen in the first scan were blurred or even disappeared in the 7th scan. For example, two small white objects (1 and 5) were blurred in

the last scan. One tiny object (4) and two elongated structures (2 and 3) disappeared in the last scan (Fig. 5). The estimated X-ray dose for each scan was about 5 MGy determined by using the program RADDOSE3D[33]. To quantify the progressive change in cell volume with respect to repeated X-ray scans and radiation damage, we measured the alternation in cell dimension as shown in Supplementary Fig. 7. Notably, changes in cell volume were observed after the first three scans, equivalent to an X-ray dose of 15 MGy. We suggest that for single-cell X-ray tomography, the total dose should be limited to 15 MGy or lower.

## Discussion

In this work, we developed an experimental and computational workflow that allows for the correlation of two distinct bioimaging modalities: nano-XCT and nano-XRF imaging. This approach enables the investigation of intracellular structures and chemical-element distributions within the same cells. The fast XCT serves as an effective screening tool to identify cells with interesting structural features. The selected cells are then scanned with XRF to obtain elemental maps, providing complementary information about the distribution of specific elements within the cell. In addition, the tomogram of a cell serves as a spatial framework, allowing for the precise localization of elemental maps within the context of cellular structure. The correlation between X-ray computed tomography and fluorescence provides valuable insights into cellular compositions (Fig. 4). For instance, phosphorus, a key component of DNA and chromatin, enriches the central region of the cell, corresponding to the segmented nucleus. Interestingly, zinc, which exhibits stronger fluorescence signals compared to phosphorus, co-localized with phosphorus in the same cellular region. This suggests a potential involvement of zinc in DNA-related biological activities, supported by previous research[34].

The abundance of each chemical element within a cell has a wide range of distributions, resulting in XRF signals with varying intensities. Further optimization of detector sensitivity and data collection conditions may increase the signal-to-noise ratios for better-detecting low-concentration trace elements such as Fe, Mn, Co, and Ni, which are essential to support proper cell function[35]. During our cell culture and sample preparation, no additional chemical elements were introduced. The observed distribution and localization of chemical elements within the HEK293 cell likely reflect their physiological concentrations and localization. In addition to correlating two X-ray imaging modalities, further correlation with fluorescence optical microscopy will provide additional insights into the function of transporter proteins associated with these elements. For instance, over-expression or knockdown of zinc transporters[36,37] in the plasma membrane and intracellular membranes of a cell can shed light on their role in regulating zinc homeostasis and their physiological and pathological implications.

Radiation damage presents a challenge in achieving high-resolution imaging of biological samples[38,39]. Chemical fixation has proven to be an effective method for preserving biological samples and mitigating radiation damage[23,40]. By employing chemical fixation, it was possible to perform hard X-ray imaging on cells at room temperature while preserving the integrity of intracellular features[13,25,41,42]. To prepare radiation-resistant cell samples, we employed an extended fixation time of 2 h using PFA followed by freeze-drying to remove water. Live-cell imaging provides an ideal approach to monitor the fixation process and identify optimal fixation time for multi-modality X-ray experiments (Supplementary Movie 1). We hypothesize that the longer fixation duration allowed for better penetration of PFA into intracellular organelles, including the nucleus, for more complete fixation of cellular contents. Our sample preparation method was developed specifically for HEK293 cells, but it is generalizable and can be applied to identify optimal sample preparation conditions for other cell types. In our sample preparation, we deliberately avoided the use of contrast-enhancing metals, such as osmium, lead, and uranyl, as they have the potential to introduce additional artifacts to the cellular structure and interfere with the XRF analysis.

In other techniques, such as cryogenic X-ray crystallography and cryo-electron microscopy (cryo-EM), the recommended radiation doses for biological samples are 20 MGy for frozen-hydrated samples and 0.38 MGy for room-temperature hydrated samples[41,43]. With hydrated algae *Chlorella*, doses of 0.0015 MGy changed the ultrastructural organization. With hydrated living Chinese hamster ovarian (CHO) cells, doses of about 0.1 MGy caused morphological changes[44]. Dehydrated cells are more resistant to X-ray radiation damage than hydrated cells. It has been reported in several studies that dose damage is more visible in wet samples than in dehydrated samples[44,45]. For chemically fixed hydrated samples, doses of about 1 MGy were shown to cause cell shrinkage and mass loss[46]. In contrast, for dehydrated samples, doses of up to 10 MGy did not lead to appreciable mass loss[46]. It seems the combination of dehydration and chemical fixation will increase the dose damage resistance. In our study, we investigated the dose damage effects in dehydrated and chemically fixed cells. We believe that the dose damage delivered by one XCT (5 MGy) is consistent with the nanometer resolution features without observing any visible structural changes. Remarkably, we observed no visible overall cell volume shrinkage even after three scans, resulting in an accumulated dose of 15 MGy, which is higher than the tolerable dose (1 MGy) for hydrated cells. In our time-dependent fixation test, we found that 2-h fixation (Fig. 2) is better than 15-min and 60-min fixation (Supplementary Fig. 2) in terms of preserving structural features characterized by XCT analyses. A long fixation time in the sample preparation is thus recommended for XCT measurements to alleviate the radiation damage challenge. Exploring the precise relationship between fixation time and X-ray damage could be an intriguing avenue for future studies.

In our multimodal experiments, we conducted XRF measurements after performing seven scans for XCT data collection. However, we didn't consider the cumulative dose prior to the XRF experiments. Consequently, we observed a slight discrepancy between the XRF maps obtained after the seventh scan and the X-ray computed tomography data from the initial scan, attributed to cell shrinkage and radiation damage (Fig. 5 and Supplementary Fig. 7). XRF imaging primarily detects chemical elements bound to proteins and nucleic acids, making it less sensitive to radiation damage compared to fine cellular structures. Retrospectively, a more rational experimental strategy would have been to allocate the total dose of 15 MGy between XCT and XRF data acquisition for better results. Nevertheless, our analysis of radiation damage underscores the significance of careful planning and the selection of appropriate experimental conditions for correlative multimodal X-ray experiments.

## Methods
### Visualization of the chemically fixed cell using live-cell imaging
Adherent HEK293 cells (ATCC Cat# CRL-1573) were cultured in Dulbecco's Modified Eagle's Medium (DMEM) supplemented by 10% (v/v) fetal bovine serum and 100 units/ml of penicillin–streptomycin (Cat# 15140122, Thermo Fisher). Cell subculture was performed twice a week to maintain cell viability and proliferation. To visualize the cell growth, 1 ml of suspension of cells with a concentration of $1 \times 10^5$ cells/ml was seeded onto a 35 mm Ibidi μ-Dish (Cat#80136, Ibidi) and allowed cells to grow overnight. Cell growth was visually inspected by a light microscope before chemical fixation.

To visualize the chemical fixation process, a 4% (v/v) paraformaldehyde (PFA) solution was diluted from 16% stock (Cat# 50-980-487, Electron Microscopy Sciences) by 1× Gibco™ DPBS (Cat# 14190-136, Thermo Fisher). Firstly, cell culture media was discarded carefully, followed by two washes, each with 1 ml of 1× Gibco™ DPBS. Subsequently, the fixation solution was gently added from the side of the well to cover the middle area of the dish. A transparent lid was placed on the dish, and the dish was placed in stage top $CO_2$ incubator on a 3D Cell Explorer-fluo light microscope (Nanolive). The incubator was operated at 37 °C, 5% $CO_2$, and 95–100% humidity. The microscope was operated to perform real-time imaging for 2 h, with images captured at 30-s intervals.

## Cell culture on $Si_3N_4$ membrane

Seeding cells on $Si_3N_4$ membranes were adapted from the reported procedure[21]. $Si_3N_4$ membranes (Cat# NX5200E, Norcada) were glow-discharged using a PELCO easiGlow Glow Discharge Cleaning System (Tedpella) operated at 30 mA for 30 s. Subsequently, they were treated with 10 μl of poly-L-lysine (Cat# 25988-63-0, Sigma) and incubated in a cell-culture incubator (37 °C and 8% $CO_2$) for 25 min. Afterward, the membranes were washed 3 times for 10 s each with sterile MilliQ water and vacuum-dried overnight. Then, the membranes were seeded with 10 μl of $1 \times 10^6$ cells/ml in a Falcon 24-well cell culture plate (Cat# 353047, Fisher Scientific), and the plate was incubated in the cell-culture incubator (37 °C and 8% $CO_2$) for 25 min, resulting in roughly 250 cells/mm² on the $Si_3N_4$ after freeze-drying. Furthermore, 500 μl of the cell culture media was gently added from the side of the well to ensure the coverage of the entire membranes. The plate was then stored in the incubator for overnight cell culture.

## Chemical fixation

The chemical fixation procedure was modified from[21,25]. Cell culture media was removed from the wells containing the $Si_3N_4$ membrane with attached cells. Subsequently, the $Si_3N_4$ membranes were washed with 1 ml DPBS twice and fixed for 2 h with 1 ml of 4% PFA as previously described. After fixation, membranes were washed twice with 1 ml of ammonium acetate (150 mM, pH 7.4) to remove residual fixatives.

## Plunge-freezing and freeze-drying

To prepare the samples for freeze-drying, samples were blotted manually by using filter paper to remove liquid on the frame and the back of the $Si_3N_4$ window. The plunge-freezing was performed using a Mark IV Vitrobot (Thermo Fisher). Frozen samples were then transferred into a pre-cooled 2-ml screw cap tube (Cat# HS10060, Millipore Sigma) for freeze-drying using a FreeZone Console Freeze Dryer (Cat# 711211110, Labconco). After the freeze dryer reached −80 °C with a vacuum level of 0.05 mbar, the sample tubes were quickly transferred into a freeze-drying container preloaded with liquid nitrogen. The container was swiftly connected to the freeze-drying machine while the vacuum valve was turned from off to on. The samples were subjected to freeze-drying for 2 days to ensure complete removal of water.

## Cell imaging and labeling for correlated multimodality data acquisition

To collect correlative data at multiple X-ray beamlines, it is necessary to perform cell imaging and labeling. We used a Leica TCS SP5 (Leica Microsystems Inc., Deerfield, IL, USA) to provide an overall view of cells on the $Si_3N_4$ membranes as well as to label cells for correlated X-ray imaging data acquisition. The freeze-dried sample was gently put upside down on the center of a μ-Dish before imaging. A 10× objective lens was employed to survey the cells present on the membranes. Once a specific region of interest was identified, the objective lens was switched to 50x magnification to capture high-resolution images for cell labeling, serving as a reference for subsequent X-ray data collection.

## X-ray computed tomography data collection

The samples were manually mounted onto the FXI beamline data collection stage, and the target cell was identified using a telescope that precisely aligned with the X-ray beam. The XCT data were acquired using X-rays at 7 keV. Two different exposure schemes were employed: 0.1 s exposure at a speed of 3°/sec and 0.2 s exposure at a speed of 1.5°/s. For each scheme, three data sets were collected with a defocus of 0, 50, and 100 μm. To assess the extent of radiation damage, an additional data set was collected with the same condition as used for the first data set.

## Correlated XRF data collection

The same sample was used at the SRX beamline[47] for XRF measurements following the XCT experiments. At SRX, a pair of rhodium-coated KB (Kirkpatrick–Baez) mirrors was used to focus the X-ray beam down to 190 nm (H) × 230 nm (V) nm. By employing an in-line optical light microscope, we were able to locate the same cell for correlative chemical mapping. The focused beam was used to raster scan the cell, allowing for the measurement of elements of biological significance. Cells were fly scanned with 100 nm step size with 0.1 s dwell time, with the photon energy at 12 keV and flux of about $4.5 \times 10^{11}$ ph/s. The estimated dose is 14 MGy per pixel. These XRF maps are scanned in both horizontal and vertical directions with the same steps. For the 0° scan, the step size was 100 nm, with $270 \times 301$ pixels. For the other angles, the step size was 300 nm, with $141 \times 101$ pixels. A Vortex 7-element silicon drift detector (Hitachi, Ltd.) was located at an angle of 75° with respect to the incident beam to capture the fluorescence emission. A collimator was put before the detector with a large opening next to the sample to optimize the fluorescence signals. The obtained spectra were subsequently analyzed using the open software PyXRF[48] to generate elemental distribution maps for each element.

## X-ray imaging data analysis

X-ray tomograms were reconstructed using TXM-Sandbox[27] and IMOD[28]. As described in TXM-sandbox, the rotation center position was first identified in TXM-sandbox's "Trial Center" mode in the raw images. Three filters from the TXM sandbox were used to enhance the signal-to-noise ratio (SNR). Initially, a stripe removal filter with a pixel size range of 21–81 was used to minimize ring artifacts. Subsequently, a phase retrieval filter and a median filter with a kernel size of $5 \times 5$ were applied to further improve the SNR. The choice of filter parameters was guided by the SNR of the reconstructed image. Then a full image reconstruction was performed with "Gridrec" algorithm under "Volume Reconstruction" mode. For the reconstruction, the voxel size is 21 nm, and the field of view is 52 μm.

The cell membrane was segmented semiautomatically using FIJI[49] (https://imagej.net/software/fiji/) and IMOD. First, a Canny edge detector was employed to enhance the boundaries within a predefined region of interest in the image, followed by the application of thresholding to convert these enhanced boundaries into binary masks. The threshold value was chosen to retain only the cell membrane boundaries while eliminating the nucleus membrane boundaries. Afterward, image binary opening and closing operations were conducted to eliminate smaller noise objects. First, a Canny edge detector was used to enhance the cell membrane region and then thresholding was used to convert the enhanced membrane region to binary masks. The resultant membrane masks were converted to a 3D IMOD model using "imodmop" and "imodauto" commands. The cell nucleus was segmented manually using IMOD. The resultant 3D model was visualized with IMOD and Napari[50] (http://narpari.org).

## Correlation of X-ray computed tomography and fluorescence images

The reconstructed 3D X-ray tomograms were first projected into 2D projection maps along the 4 different angles (−15°, 0°, 25°, 55°). Then 2D rigid image registration (translation + rotation) was performed with FIJI[49] (https://imagej.net/software/fiji/) on 2D X-ray projection image and 2D fluorescent image separately for four different angles. The registered images from two imaging modalities were overlaid and visualized with Napari.

## Reporting summary

Further information on research design is available in the Nature Portfolio Reporting Summary linked to this article.

## Data availability

All data are available from the corresponding authors upon reasonable request. Image data were deposited to figshare and can be accessed via https://doi.org/10.6084/m9.figshare.25234129.v1.

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

## Acknowledgements

This work was supported by Laboratory Directed Research and Development Program LDRD21-013 of Brookhaven National Laboratory and the U.S. Department of Energy (DOE), Office of Biological and Environmental Research (KP1601011). The work used the Full-field X-ray Imaging (FXI) beamline at 18-ID and the Submicron Resolution X-ray Spectroscopy (SRX) beamline at 5-ID of the National Synchrotron Light Source II, a U.S. DOE Office of Science User Facility operated for the DOE Office of Science by Brookhaven National Laboratory under Contract No. DE-SC0012704. The work used the Vitrobot at the Laboratory for BioMolecular Structure, which is supported by the U.S. DOE Office of Biological and Environmental Research (KP1607011). The work used the confocal microscope at the Center of Functional Nanomaterials, which is a U.S. DOE Office of Science User Facility, at Brookhaven National Laboratory under Contract No. DE-SC0012704.

## Author contributions

Y.Y., X.X. and Q.L. designed the study and experiments. Z.L., X.Z., Y.Y. and X.X. performed the experiments. Z.L., X.Z., P.N., Y.L., L.W., Y.S.C., T.P., Y.Y., X.X. and Q.L. analyzed the data. Z.L. Y.Y., X.X. and Q.L. wrote the paper with help from other coauthors.

## Competing interests

The authors declare no competing interests.
