## [Peer review file · Communications Biology]

Reviewers' comments:

Reviewer #1 (Remarks to the Author):

Communications biology - Nature

Statement to the authors

Manuscript ID: COMMSBIO-23-3122-T

Title of the work: Correlative single-cell X-ray tomography and X-ray fluorescence imaging

The presented study describes an integrated X-ray nano-tomographic and high-resolution X-ray fluorescence imaging study on single cells utilizing synchrotron beamlines. The steps described comprise sample preparation, the execution of the respective imaging measurements, data processing, as well as image registration and analysis.

In terms of sample preparation chemical fixation and freeze drying, as well as cell targeting and labelling in confocal microscopy (Leica TCS SP5). The synchrotron experiments were performed at the National Synchrotron Light Source II (NSLS-II) at Brookhaven National Laboratory (BNL). Transmission tomography was performed at the Full-field X-ray Imaging (FXI) beamline, whereas X-ray fluorescence imaging was performed at the Resolution X-ray Spectroscopy (SRX) beamline, both at room temperature. Following reconstruction and segmentation of the tomographic data, fluorescence data taken at four different rotation angles were mapped onto 2d maximum intensity projections of the tomographic data in the corresponding volume orientations. Such superimpositions are shown for Ca, Cu and Zn (in the supplementary material also for Cl, Fe and P). Although authors describe a cell shrinkage during the measurements

The overall quality of the manuscript appears adequate. The structure of the document is clear and the provided figures are relevant and of adequate visual quality. There is a comprehensive description of the scientific background. However, there are occasional spelling errors unclear formulations.

To me the topic appears interesting in general, however there are several shortcomings in the presented study that in my opinion contradict publication, at least in the current form of the manuscript.

Points of critique:

1: The presented work lacks scientific focus and offers a limited level of innovation. The imaging capabilities of the employed beamlines are established, and so is the majority of the employed equipment and methods (confocal microscopy, employed software tools (TXM sandbox, IMOD, RADDSE3D, PyXRF, FIJI,...)). The projection, alignment and superimposition of the CT and XRF data

on the other hand is rather straightforward. For this reason, it appears the main point of innovation in this study is the sample preparation and its effect on the sample behavior, although this is not my field of expertise. Therefore I would expect a rigorous study of the corresponding process parameters and their impact on the sample behavior in terms of radiation hardness, longitudinal stability, etc.. However, there is little to be learned here. For instance, although different fixation durations for the cells apparently were tested,

line121ff: "Subsequently, PFA-fixed cells with fixation durations of 15 min, 60 min, and 120 min were subjected to vitrification and freeze-drying",

the sole sample shown in this study has been subjected to 120 minutes of fixation with the following reasoning:

line 258ff: "To prepare radiation-resistant cell samples, we employed an extended fixation time of 2 hours using paraformaldehyde (PFA) followed by freeze-drying to remove water. In our initial tests, insufficient PFA fixation (15 minutes) resulted in bubbles or the loss of internal features during XRT analysis"

Further:

line 276ff: "Nonetheless, a long fixation time in the sample preparation is recommended for XRT measurements to alleviate the radiation damage challenge. Exploring the precise relationship between fixation time and X-ray damage could be an intriguing avenue for future studies."

Here the authors imply a monotonic dependency of radiation sensitivity to the fixation time without any data to support this claim. Indeed, as the authors suggest, investigating this dependency would have been of relevance for this study.

Also, the outlook given by the authors does not reveal major innovation. For instance, to enhance tomographic contrast, the authors propose to use phase contrast tomography instead of attenuation in a next step (line 299). This is a rather obvious continuation, which, however, undeniably could have a positive impact on feature contrast and discriminability.

2: An apparent logical inconsistency arises from the following statement:

Line 280ff: "In our multimodal experiments, we conducted XRF measurements after performing seven scans for XRT data collection. However, we didn't consider the cumulative dose prior to the XRF experiments. Consequently, we observed a slight discrepancy between the XRF maps obtained after the seventh scan and the X-ray tomography data from the initial scan, attributed to cell shrinkage and radiation damage"

I suppose this statement implies that the authors used the initial dataset for the superimposition with the XRF imaging. In this case to me it is unclear why the authors did not utilize the last tomographic dataset for the superimposition, thus minimizing this discrepancy.

3: The term "optimization" is used several times in an unprecise manner:

A: Line 110ff: "the fixation conditions were optimized using 4% paraformaldehyde (PFA), a routinely employed fixative for cell fixation before immunolabeling and x-ray fluorescence imaging ". However, there is no parameter variation mentioned.

B: line 123ff: "the cell density on the Si₃N₄ membrane was optimized to ensure sufficient separation between cells". What was the target value?

C: line 385f: "Various filter parameters, reconstruction algorithms and parameters were tested under this mode to find the optimal reconstruction condition". How? What was used in the end?

D: line 137ff: "Taking into account the optimal imaging contrast and the flux, we conducted all the tomography experiments at 7 keV". What exactly was taken into account and how?

If the authors speak of optimization this should entail a description of the optimization target and the methods employed.

4: An interesting point of the presented study is the apparently low radiation sensitivity of the sample at room temperature. Here a more rigorous investigation could provide valuable insights into how this particular approach of sample preparation affects its radiation hardness.

Line 270ff: "Remarkably, we observed no significant overall cell volume shrinkage even after three scans, resulting in an accumulated dose of 15 MGy, which is significantly higher than that used for hydrated cells."

This must be put into context. What cell shrinkage is expected for hydrated cells? Is there any reference data?

5: Line 217:). "In addition, high-resolution features seen in the first scan were blurred or even disappeared in the 7th scan".

It would be good to discuss the described alterations explicitly with reference to the corresponding figures.

6: line 273ff: "Since the spatial resolution in XRT is about two orders of magnitude lower than in X-ray crystallography and cryo-EM, it is reasonable to assume the tolerance to X-ray dose in XRT is higher than the thresholds used in X-ray crystallography and cryo-EM."

I'm not sure what you want to imply with this sentence. What do you mean by tolerance? The signal generation mechanisms for the mentioned modalities differ strongly. Thus, a straightforward extrapolation of the dose requirements based on spatial resolution seems a bit far-fetched.

Reviewer #2 (Remarks to the Author):

In this work, the authors developed a single-cell X-ray imaging workflow that encompasses sample preparation, targeting, multi-modality nano-XRT and nano-XRF imaging data acquisition, and computational alignments of multi-modality image data at nanometer resolutions. This approach

enables the investigation of intracellular structures and chemical-element distributions within the same cells. The fast XRT serves as an effective screening tool to identify cells with interesting structural features. The selected cells are then scanned with XRF to obtain elemental maps, providing complementary information about the distribution of specific elements within the cell. The results of the research pave the way for the broad application of hard X-rays for correlative characterization of cell structure and function.

In general, the investigated topic is very interesting and the experimental approach is justifiable. I think this manuscript has solid contributions but needs some revisions before publication.

1. Line 141: The term “volume rendering” may cause ambiguity, maybe here refers to the projection?
2. Line 208: Please provide the estimated X-ray dose of the XRF imaging.
3. Line 361: What is the reconstruction voxel size and the field of view of the X-ray tomography imaging?
4. Line 369: What is the number of horizontal and vertical scanning steps in XRF imaging?
5. X-ray fluorescence imaging is a quantitative imaging method. Quantitative elements mapping holds enhanced value for scientific research. Have the authors conducted experiments pertaining to quantitative X-ray fluorescence imaging?

Reviewer #1 (Remarks to the Author):

Communications biology - Nature
Statement to the authors

Manuscript ID: COMMSBIO-23-3122-T

Title of the work: Correlative single-cell X-ray tomography and X-ray fluorescence imaging

1: The presented work lacks scientific focus and offers a limited level of innovation. The imaging capabilities of the employed beamlines are established, and so is the majority of the employed equipment and methods (confocal microscopy, employed software tools (TXM sandbox, IMOD, RADDPOSE3D, PyXRF, FIJI,...)). The projection, alignment and superimposition of the CT and XRF data on the other hand is rather straightforward. For this reason, it appears the main point of innovation in this study is the sample preparation and its effect on the sample behavior, although this is not my field of expertise. Therefore, I would expect a rigorous study of the corresponding process parameters and their impact on the sample behavior in terms of radiation hardness, longitudinal stability, etc. However, there is little to be learned here.

Response: We appreciate the reviewer's comment on the lack of scientific focus and the level of innovation. X-ray tomography (XRT) is a full-field imaging tool to reveal 3D cell structure. X-ray scanning fluorescence (XRF) is a spectroscopic imaging tool for simultaneous mapping of the abundance and localization of multiple chemical elements. Both imaging modalities have been well established as pointed out by the reviewer. However, to the best knowledge of the authors, there is no available method to make correlative studies to perform XRT and XRF experiments to get the structure and chemical information from the same cell. To fill this technical gap, the scientific innovation of our work is to combine XRT and XRF in the way that XRT can be used for high-throughput identification of cells for more detailed chemical mapping using XRF. In addition, our work is the first example of using hard X-rays for cellular tomography without using heavy-metal staining. We revised the title of the manuscript to emphasis on "hard" x-ray tomography "Correlative single-cell hard X-ray tomography and X-ray fluorescence imaging". The scientific interest of our work is echoed by the reviewer#2 "The investigated topic is very interesting and the experimental approach is justifiable".

We thank the reviewer for the recognition of our novelty in sample preparation. In our revised manuscript, we have added additional experiments and analyses regarding process parameters, radiation hardness, and longitudinal stability.

For instance, although different fixation durations for the cells apparently were tested, line121ff: "Subsequently, PFA-fixed cells with fixation durations of 15 min, 60 min, and 120 min were subjected to vitrification and freeze-drying", the sole sample shown in this study has been subjected to 120 minutes of fixation with the following reasoning: line 258ff: "To prepare radiation-resistant cell samples, we employed an extended fixation

time of 2 hours using paraformaldehyde (PFA) followed by freeze-drying to remove water. In our initial tests, insufficient PFA fixation (15 minutes) resulted in bubbles or the loss of internal features during XRT analysis"

Further:

line 276ff: "Nonetheless, a long fixation time in the sample preparation is recommended for XRT measurements to alleviate the radiation damage challenge. Exploring the precise relationship between fixation time and X-ray damage could be an intriguing avenue for future studies."

Here the authors imply a monotonic dependency of radiation sensitivity to the fixation time without any data to support this claim. Indeed, as the authors suggest, investigating this dependency would have been of relevance for this study.

Response: We thank the reviewer for this great suggestion. We have performed a study to study the relationship between fixation time and X-ray damage. In the revised manuscript, we added a Supplementary Fig. 2 (see figure below) to show the XRT results to explore the dependency of radiation sensitivity to the fixation time (0 min, 15 min and 60 min). Without fixation, the cell collapsed on the Si₃N₄ membrane with radiation-induced bubbles. With a fixation time of 15 mins and 60 mins, the cells preserved the 3D volume, but still showed bubbles or smeared internal features. We added the following text to the revised manuscript (**page 5, lines 121-127**).

"Chemical fixation of cells is essential to maintain cellular structures for XRT analysis. Insufficient PFA fixation (0 minutes and 15 minutes) resulted in bubbles and the loss of internal features during XRT analysis (Supplementary Fig. 2a, b). Without fixation (0 minutes), the freeze-dried cell collapsed and was flatted on the Si₃N₄ membrane (Supplementary Fig. 2a). Sixty-minute fixation did not result in bubbles, but the internal cellular features were mostly lost compared to those of the 120-minute fixation (Supplementary Fig. 2c and Fig. 2c-d)."

Supplementary Fig. 2. XRT reconstructions of insufficiently fixed cells. The projections of XRT tomograms at two orthogonal views. Arrows indicate bubbles induced by X-ray exposure. Scale bar is 5 μm . **a**, no fixation (0 minutes). **b**, 15 minutes of fixation. **c**, 60 minutes of fixation.

Also, the outlook given by the authors does not reveal major innovation. For instance, to enhance tomographic contrast, the authors propose to use phase contrast tomography instead of attenuation in a next step (line 299). This is a rather obvious continuation, which, however, undeniably could have a positive impact on feature contrast and discriminability.

Response: We thank the reviewer for pointing out the contrast enhancement technique. We acknowledge that using phase contrast tomography undeniably has a positive impact on feature contrast and discriminability. To reduce confusion, we have removed this obvious outlook from the revised manuscript.

An apparent logical inconsistency arises from the following statement:

Line 280ff: “In our multimodal experiments, we conducted XRF measurements after performing seven scans for XRT data collection. However, we didn’t consider the cumulative dose prior to the XRF experiments. Consequently, we observed a slight discrepancy between the XRF maps obtained after the seventh scan and the X-ray tomography data from the initial scan, attributed to cell shrinkage and radiation damage”

I suppose this statement implies that the authors used the initial dataset for the superimposition with the XRF imaging. In this case to me it is unclear why the authors did not utilize the last tomographic dataset for the superimposition, thus minimizing this discrepancy.

Response: We thank the reviewer for this suggestion. We now included a supplemental figure (**Supplementary Figure 5b**) to show the superimposition of XRF data with the XRT data from the last scan (i.e. 7th scan). However, the last scan XRT had shown significant internal feature changes caused by radiation damage (**Fig. 5**). Considering that only a 5% longitudinal change from radiation damage, our superimposition and analysis on the XRF mapping using the first scan remain valid (**Fig. 4**).

Supplementary Fig. 5.

3: The term "optimization" is used several times in an unprecise manner:

A: Line 110ff: "the fixation conditions were optimized using 4% paraformaldehyde (PFA), a routinely employed fixative for cell fixation before immunolabeling and x-ray fluorescence imaging ". However, there is no parameter variation mentioned.

Response: We optimized the time for the PFA fixation. We tested different fixation times and found that 120 min fixation gave a better result than 0 minutes, 15 minutes, and 60 minutes of fixation time. We have revised the text to make this clear (**page 5, lines 121-127**).

B: line 123ff: "the cell density on the Si₃N₄ membrane was optimized to ensure sufficient separation between cells". What was the target value?

Response: During the angular rotation (XRF and XRT), we attempted to eliminate overlapping cells which may complicate the angular XRF and XRT experiments. We thus used this criterion and tested different concentrations of cells. We found that Si₃N₄ membranes seeded with 10 μl of 1x10⁶ cells/ml gave monolayer separation before fixation, which is twice as much as the protocol from reference 21. After fixation and freeze-drying, Si₃N₄ membranes had reasonable separation of cells (roughly 250 cells/mm²).

We've revised the method section with text below (**page 12, lines 337-340**):

"Then, the membranes were seeded with 10 μl of 1x10⁶ cells/ml in a Falcon 24-well cell culture plate (Cat# 353047, Fisher Scientific) and the plate was incubated in the cell-culture incubator (37 °C and 8% CO₂) for 25 minutes, resulting in roughly 250 cells/mm² on the Si₃N₄ after freeze-drying."

C: line 385f: "Various filter parameters, reconstruction algorithms and parameters were tested under this mode to find the optimal reconstruction condition". How? What was used in the end?

Response: We have added detailed information about filters and reconstruction algorithms in the revised manuscript (**page 14, lines 395-401**).

"Three filters from the TXM-sandbox were used to enhance the signal-to-noise ratio (SNR). Initially, a stripe removal filter with a pixel size range of 21-81 was used to minimize ring artifacts. Subsequently, a phase retrieval filter and a median filter with a kernel size of 5x5 were applied to further improve the SNR. The choice of filter parameters was guided by the SNR of the reconstructed image. Then a full image reconstruction was performed with "Gridrec" algorithm under "Volume Reconstruction" mode."

D: line 137ff: "Taking into account the optimal imaging contrast and the flux, we conducted all the tomography experiments at 7 keV". What exactly was taken into account and how? If the authors speak of optimization this should entail a description of the optimization target and the methods employed.

Response: Based on the physical interactions between X-rays and matter, the diffraction intensity is proportional to the cubic X-ray wavelength. Lower energy in general increases intensity and contrast. With the optics set up at the FXI beamline at NSLS-II, 7 keV is the lowest X-ray energy we can reach. Therefore, we used 7 keV for our XRT experiments. We added two sentences to rationalize our consideration of the use of X-ray energies for our XRT experiments (**page 5, lines 143-144**).

“Lower energy X-rays increase intensity and contrast. With the optics set up at the FXI beamline at NSLS-II, 7 keV is the lowest X-ray energy we can reach.”

4: An interesting point of the presented study is the apparently low radiation sensitivity of the sample at room temperature. Here a more rigorous investigation could provide valuable insights into how this particular approach of sample preparation affects its radiation hardness.

Line 270ff: “Remarkably, we observed no significant overall cell volume shrinkage even after three scans, resulting in an accumulated dose of 15 MGy, which is significantly higher than that used for hydrated cells.”

This must be put into context. What cell shrinkage is expected for hydrated cells? Is there any reference data?

Response: We appreciate the reviewer’s suggestion for further investigation, analysis, and reference data. We have added the below text to the revised manuscript (**pages 10-11, lines 279-290**).

“With hydrated algae *Chlorella*, doses of 0.0015 MGy changed the ultrastructural organization. With hydrated living Chinese hamster ovarian (CHO) cells, doses of about 0.1 MGy caused morphological changes⁴⁴. Dehydrated cells are more resistant to X-ray radiation damage than hydrated cells. It has been reported in several studies that dose damage is significantly more visible in wet samples than in dehydrated samples^{44,45}. For chemically fixed hydrated samples, doses of about 1 MGy were shown to cause cell shrinkage and mass loss. In contrast, for dehydrated samples, doses of up to 10 MGy did not lead to appreciable mass loss⁴⁶. It seems the combination of dehydration and chemical fixation will increase the dose damage resistance. In our study, we investigated the dose damage effects in dehydrated and chemically fixed cells. We believe that the dose damage delivered by one XRT scan (5 MGy) is consistent with the nanometer resolution features without observing any visible structural changes.”

(page 11, lines 293-295)

“In our time-dependent fixation test, we found that 2-hour fixation (Fig. 2) is better than 15-min and 60-min fixation (Supplementary Fig. 2) in terms of preserving structural features characterized by XRT analyses.”

5: Line 217: “In addition, high-resolution features seen in the first scan were blurred or even disappeared in the 7th scan”.

It would be good to discuss the described alterations explicitly with reference to the corresponding figures.

Response: We thank the reviewer for this great suggestion. We added two blue sentences to describe alterations explicitly (**page 8, lines 222-224**).

“For example, two small white objects (1 and 2) were blurred in the last scan. One tiny object (4) and two elongated structures (2 and 3) disappeared in the last scan (Fig. 5).”

6: line 273ff: "Since the spatial resolution in XRT is about two orders of magnitude lower than in X-ray crystallography and cryo-EM, it is reasonable to assume the tolerance to X-ray dose in XRT is higher than the thresholds used in X-ray crystallography and cryo-EM."

I'm not sure what you want to imply with this sentence. What do you mean by tolerance? The signal generation mechanisms for the mentioned modalities differ strongly. Thus, a straightforward extrapolation of the dose requirements based on spatial resolution seems a bit far-fetched. (Chris Jacobson paper)

Response: To see structural changes at a large scale, a high X-ray dose is needed. The “tolerance” here means the sample can be exposed to a higher X-ray dose without mass loss and the creation of artifacts. To reduce the confusion by tolerance and the sentence, we deleted the sentence from the revised manuscript.

Reviewer #2 (Remarks to the Author):

In this work, the authors developed a single-cell X-ray imaging workflow that encompasses sample preparation, targeting, multi-modality nano-XRT and nano-XRF imaging data acquisition, and computational alignments of multi-modality image data at nanometer resolutions. This approach enables the investigation of intracellular structures and chemical-element distributions within the same cells. The fast XRT serves as an effective screening tool to identify cells with interesting structural features. The selected cells are then scanned with XRF to obtain elemental maps, providing complementary information about the distribution of specific elements within the cell. The results of the research pave the way for the broad application of hard X-rays for correlative characterization of cell structure and function.

In general, the investigated topic is very interesting and the experimental approach is justifiable. I think this manuscript has solid contributions but needs some revisions before publication.

Response: We thank the reviewer for the positive evaluation of our work.

1. Line 141: The term “volume rendering” may cause ambiguity, maybe here refers to the projection?

Response: We thank the reviewer for the good suggestion. We changed “volume rendering” to “projection” (page 7, line 147).

2. Line 208: Please provide the estimated X-ray dose of the XRF imaging.

Response: The estimated X-ray dose for the XRF imaging is 14 MGy per pixel. We have added this dose information to the method (page 14, line 385).

3. Line 361: What is the reconstruction voxel size and the field of view of the X-ray tomography imaging?

Response: We thank the reviewer for this good suggestion. In the revised manuscript, we added the voxel size (21 nm) and the field of view (52 μm) used for the XRT imaging (page 7, lines 401-402).

“For the reconstruction, the voxel size is 21 nm and the field of view is 52 μm .”

4. Line 369: What is the number of horizontal and vertical scanning steps in XRF imaging?

Response: These XRF maps are scanned in both horizontal and vertical directions with the same steps. For the 0-degree scan, the step size was 100 nm, with 270 x 301 pixels. For the other angles, the step size was 300 nm, with 141 x101 pixels. We added such information in the methods (page 14, lines 386-388).

5. X-ray fluorescence imaging is a quantitative imaging method. Quantitative elements mapping holds enhanced value for scientific research. Have the authors conducted experiments pertaining to quantitative X-ray fluorescence imaging?

Response: We thank the reviewer’s suggestion. We did have quantitative XRF mapping capability. The work presented in the paper, however, is not related to quantifying the metal concentrations in the cells. Thus, quantification experiments using stand elements were not included during our experimentation. We will incorporate them into our future study.

REVIEWERS' COMMENTS:

Reviewer #1 (Remarks to the Author):

In the revised version of the paper the authors have addressed the original concerns issued by the reviewer(s) comprehensively.

I therefore recommend publication of the work in its current form.

Reviewer #2 (Remarks to the Author):

The paper introduces a method of using synchrotron radiation light source to perform XCT and XFCT imaging on the same cell separately. In fact, XCT and XFCT can be achieved simultaneously in a single X-ray scan, which will significantly reduce radiation dose and the damage to cells. It will also improve the applicability of this technology for cell imaging.

1. Several terms in the paper do not quite conform to the customary usage in the field of imaging. Such as X-ray tomography (XRT), should be X-ray computed tomography (XCT); X-ray fluorescence (XRF) imaging should be X-ray fluorescence computed tomography (XFCT) imaging.

2. Figure 1 lacks necessary explanation for the components in XCT and XRF imaging.

3. Line 144, the paper mentions that the minimum energy of the light source used is 7 keV. Is this energy too high for imaging a single cell?

4. Why used 12keV for XFCT? And, What are the energies of XRF photons of Ca, P, Fe, Zn, etc.

5. XRF imaging naturally contains a large number of scattered photons. The results shown in Figure 3 also illustrate this issue. Elements mappings of Ca, Fe, P, Zn appear to be widely distributed. They are likely background scattering noise, which affects the accuracy of the distributions of these elements. XRF quantitative imaging requires attenuation correction and scattering correction. Please introduce how attenuation and scattering correction are performed.

6. Line 377, in XRF data collection, was collimator used before the Vortex 7-element silicon drift detector? Why use different step size at different angles?

REVIEWERS' COMMENTS:

Reviewer #1 (Remarks to the Author):

In the revised version of the paper the authors have addressed the original concerns issued by the reviewer(s) comprehensively.

I therefore recommend publication of the work in its current form.

Response: We thank the reviewer for helping us improve the work.

Reviewer #2 (Remarks to the Author):

The paper introduces a method of using synchrotron radiation light source to perform XCT and XFCT imaging on the same cell separately. In fact, XCT and XFCT can be achieved simultaneously in a single X-ray scan, which will significantly reduce radiation dose and the damage to cells. It will also improve the applicability of this technology for cell imaging.

Response: We thank the reviewer for the comment.

1. Several terms in the paper do not quite conform to the customary usage in the field of imaging. Such as X-ray tomography (XRT), should be X-ray computed tomography (XCT); X-ray fluorescence (XRF) imaging should be X-ray fluorescence computed tomography (XFCT) imaging.

Response: We thank the reviewer's suggestion on the terms, and we have revised our manuscript using the suggested "XCT" for X-ray computed tomography. However, for X-ray fluorescence imaging, we didn't perform computed tomography reconstruction. Instead, we rotated XCT tomogram as 2D projections and aligned the projections with 2D XRF images. Therefore, we prefer to keep "XRF".

2. Figure 1 lacks necessary explanation for the components in XCT and XRF imaging.

Response: We added XCT and XRF components to Figure 1 and its caption.

3. Line 144, the paper mentions that the minimum energy of the light source used is 7 keV. Is this energy too high for imaging a single cell?

Response: Due to optics limitations, 7 keV is the lowest energy we can use for XCT imaging. In this work, we demonstrate that hard X-rays at 7 keV can be used for imaging a single cell.

4. Why used 12keV for XFCT? And, What are the energies of XRF photons of Ca, P, Fe, Zn, etc.

Response: In this work, 12 keV was used as a choice of exciting the K-edges of these elements of interest, as up to Zn.

5. XRF imaging naturally contains a large number of scattered photons. The results shown in Figure 3 also illustrate this issue. Elements mappings of Ca, Fe, P, Zn appear to be widely distributed. They are likely background scattering noise, which affects the accuracy of the distributions of these elements. XRF quantitative imaging requires attenuation correction and scattering correction. Please introduce how attenuation and scattering correction are performed.

Response: The XRF maps presented in this manuscript were filtered using non-linear least-squares fitting based on the Levenberg-Marquardt algorithm using PyXRF (ref 1). The algorithm takes into consideration elemental peaks, background, and the areas of scattering peaks (elastic and Compton) as fitting parameters. The strong background that appeared in Cl and P could arise from the residuals of the cell culture and wash media that contains chloride and phosphate.

The cell specimens are thin ($\sim 10 \mu\text{m}$) and were freeze-dried. Most of the space inside the cell is air-filled. Therefore, for the XRF data, we didn't apply any attenuation correction. If there's a need for this correction for our future studies of thick samples, we can apply the self-absorption algorithms developed at NSLS-II (ref 2).

1. Li, L., Yan, H., Xu, W., Yu, D., Heroux, A., Lee, W.K., Campbell, S.I. and Chu, Y.S., 2017, September. PyXRF: Python-based X-ray fluorescence analysis package. In X-Ray Nanoimaging: Instruments and Methods III (Vol. 10389, pp. 38-45). SPIE.1.
2. Ge, M., Huang, X., Yan, H., Gursoy, D., Meng, Y., Zhang, J., Ghose, S., Chiu, W.K., Brinkman, K.S. and Chu, Y.S., 2022. Three-dimensional imaging of grain boundaries via quantitative fluorescence X-ray tomography analysis. *Communications Materials*, 3(1), p.37

6. Line 377, in XRF data collection, was collimator used before the Vortex 7-element silicon drift detector? Why use different step size at different angles?

Response: We did use a collimator before the detector with a large opening next to the sample to optimize the fluorescence signals. We added such information to the revised manuscript.

We used different step sizes in different angles because we wanted to have the highest resolution XRF map of the cells at 0 degrees, while the other angles serve more to correlate subcellular chemical components with the 3D XCT map.